# Intervention of Coordination by Liaison Nurse Where Ward Staff Struggled to Establish a Therapeutic Relationship with a Patient Because of Failure to Recognize Delirium: A Case Study

**DOI:** 10.3390/healthcare10071335

**Published:** 2022-07-18

**Authors:** Yuri Nakai, Yusuke Nitta, Reiko Hashimoto

**Affiliations:** 1Faculty of Nursing, University of Kochi, 2751-1 Ike, Kochi City 780-8515, Japan; nakai_yuri@cc.u-kochi.ac.jp; 2Department of Neuropsychiatry, Kanazawa Medical University, 1-1 Uchinada, Kahoku 920-0265, Japan; reipon@kanazawa-med.ac.jp

**Keywords:** delirium, liaison nurse, misrecognition of delirium, coordination

## Abstract

In this case study, ward staff found it difficult to establish a therapeutic relationship with a patient with advanced gastric cancer because they misdiagnosed delirium as a psychogenic reaction to the cancer diagnosis. This article reports on the process and effects of intervention by a liaison nurse. The liaison nurse recognized the misdiagnosis and approached the ward staff via a psychiatrist-led team. This enabled rapid revision of the treatment policy. The liaison nurse contributed to the continuation of treatment by enabling the ward staff and patient to understand each other better and to collaborate to build a relationship and control the patient’s mental health symptoms, including attention disorder and excessive demands. The patient and family had different views on discharge because of the patient’s mental health issues. The liaison nurse encouraged the ward staff to inform the family caregiver about the patient’s medical condition, the expected future course of the disease, and likely symptoms, and provide appropriate professional services. This enabled the patient to be discharged in line with their wishes. This case highlights the role of the liaison nurse in coordinating care and helping ward staff to recognize symptoms and provide appropriate care and support for patients and their families.

## 1. Introduction

Delirium is caused by a range of factors such as the presence of predisposing conditions, including cognitive impairment, severe illness, visual impairment, drug therapy, and hospitalization [1]. When delirium develops during treatment, it interrupts the treatment and increases the risk of complications and further functional deterioration. It is associated with adverse events such as confusion and falls in hospitalized patients [2], and self-extraction of the infusion line [3]. This may mean that the length of hospital stay is extended, and labor costs and material costs for the stay are both increased [4]. This can also increase mortality [5]. The etiology of delirium is unknown, but prophylactic nursing interventions may be an efficient and cost-effective solution [6]. Careful observation and intervention by nurses are important to avoid the adverse consequences of delirium, allow patients to receive the desired treatment, and return to their daily lives.

The prevalence of delirium in cancer patients increases toward the end of life [7]. Delirium is a common psychiatric complication in cancer patients, but it is often not accurately recognized. In one study of psychiatric examinations of cancer patients, doctors were distracted by symptoms such as pain and missed the diagnosis of delirium in 46% of patients [8]. Overall, 61% of patients referred to palliative care had delirium overlooked by the primary referral team [9]. Approximately 50% of the primary teams of patients referred to liaison services for reasons other than delirium were unaware of the delirium [10]. In particular, delirium is prone to be unrecognized in younger cancer patients [11]. The symptoms of delirium are diverse and appear irregularly. Patients with delirium may also try to address or hide agitation, apathy, emotional instability, and disorientation [12]. If treatment continues without the presence of delirium being recognized, it will have less or no therapeutic effect. Misdiagnosis can also lengthen the period in which nurses have difficulty responding to the patient, which can have effects on the whole ward. The nurses and medical staff who directly respond to patients’ abuse and excessive demands are especially likely to experience serious psychological distress and stress. Nurses have the most frequent and closest contact with patients and play an important role in caring for and comforting delirium patients. It is therefore very important to be aware of and minimize their needs and stress [13]. Nurses and medical staff may also develop negative feelings toward patients with unrecognized delirium, which hinders the proper treatment and care for these patients. Early recognition of delirium, identification of possible causes, and provision of knowledgeable care will improve the quality and outcome of patient care [1,14]. If a nurse or other member of staff may have failed to recognize delirium, rapid intervention is needed to address the situation and reorient both patient and staff to appropriate treatment and care. Certified nurse specialists (CNS) in Japan are nurses who participate in clinical practice, consultation, coordination of activities, ethical management, education, and research [15]. These nurses are responsible for guiding staff in difficult situations, identifying learning needs, and highlighting the right approach [16]. They can therefore improve outcomes for patients.

In the case reported here, nurses and medical staff failed to recognize a variety of psychological symptoms caused by delirium in a gastric cancer patient. Instead, they believed that these were psychogenic reactions associated with the patient’s recent diagnosis of advanced gastric cancer. It was therefore difficult for the nurses to establish a therapeutic relationship with the patient. The purpose of this study is to report on the process and effects of coordination and intervention by the CNS who was part of the liaison team. This may help to improve best practice for delirium cases in younger cancer patients.

## 2. Case Presentation

### 2.1. Terminology

#### 2.1.1. Liaison Team

The liaison team in Japan is a team of three or more people, including a psychiatrist, a nurse with more than 3 years of experience in psychiatry, and a medical professional with more than 3 years of experience in psychiatry. Medical professionals include pharmacists, occupational therapists, mental health social workers, and certified psychologists. The liaison team treats and cares for mental health symptoms and psychological problems of hospitalized patients and their families, using specialized skills from a physical, mental, and social perspective. The team aims to improve the mental health of patients and their families. It also supports the physical and mental health of staff involved in treatment, improves their motivation to work, and prevents them from burning out [17,18].

#### 2.1.2. Certified Nurse Specialist (CNS)

A certified nurse specialist is a nurse who has completed the certified nurse specialist education course at a Japanese graduate school and has passed the certified nurse specialist certification examination. As of February 2022, there are 14 specialist fields, including cancer nursing, psychiatric mental health nursing, and community health nursing. CNSs have six key roles: practice, consultation, coordination, ethical coordination, education, and research [19].

### 2.2. Case Information

This case report describes a woman in her 40s. She works as a salesperson, has never been married, and lives alone. Her parents are dead. Her family caregiver is her younger brother. Her only medical history is an oral treatment for bronchial asthma. She has no history of psychiatric treatment.

### 2.3. Clinical Findings

In June 20XX, the patient complained of an enlarged feeling in her abdomen and was diagnosed with advanced gastric cancer (stage IV). She was admitted to medical oncology on July 18. After admission, she was informed that her prognosis was 1 year. She chose to receive chemotherapy and started anticancer drugs (TS1 + cisplatin) (SP therapy) on July 20th. The second cycle was carried out on 10 August. From 15 August, she started frequent refusal of treatment and examinations and behaved aggressively toward the ward staff. On the night of 30 August, the patient’s words and actions became uncoordinated and communication became difficult. Her dissatisfaction, anger, and shouting worsened. The patient’s symptoms and ward staff behavior by stage of medical treatment are shown below.
30 August–14 September

After August 10th, the patient was reported as developing an intimidating attitude and emotional instability, making excessive demands on the nurses, having a silly smile, and being silent and immobile. The ward staff decided that these symptoms were a psychogenic reaction to the news that the cancer was advanced. They, therefore, responded without complaint to the patient’s demands and requests.
2.15 September–10 October

The patient was prescribed an antipsychotic drug, and her anger disappeared, but the irritability remained. The psychological burden on the ward staff was heavy, and they continued to struggle to deal with the patient. The patient started to show overactivity, attention dysfunction, excessive demands, and frequent family phone calls. These seem to have been triggered by spending nights away from the ward, and the resumption of the therapy. The psychological burden on the ward staff, therefore, became even higher, and they began to complain to the head nurse that it was difficult to deal with this patient on a general ward.
3.11 October–1 November

After three cycles of SP therapy, the patient showed attention dysfunction and garrulity, but wanted to be discharged to the community and live alone. The ward staff decided that she would be able to do this by using social resources such as home-visit nursing. However, her family caregiver thought it would be difficult for her to live alone because of her symptoms. He was also worried that if the patient’s condition worsened or suddenly changed, treatment would be delayed. There was therefore a conflict of opinion about discharge between the patient and her family caregiver. The ward staff respected the views of the family caregiver rather than those of the patient and suggested that the patient should give up the idea of living alone.

### 2.4. Timeline

The phase-by-phase medical conditions and the liaison nurse’s interventions are shown below.

The need for the liaison nurse to intervene, and the nurse’s actions, were classified into Phases 1, 2, and 3 by treatment and date. Figure 1 shows the patient’s physical and psychotic symptoms, blood test values, and treatments by phase.

### 2.5. Therapeutic Interventions

#### 2.5.1. Interventions of the Liaison Nurse with Ward Staff: Phase 1

##### Coordination by Liaison Nurse in Phase 1

The liaison nurse held a conference with the liaison team and ward staff to correct the failure to recognize delirium. At the conference, the liaison nurse explained that the patient’s psychological symptoms may be caused by delirium related to the chemotherapy, rather than a psychogenic reaction to her diagnosis. The team then clarified the role of each group of staff in supporting patients with delirium, to help to rebuild the support relationship between the patient and ward staff. Physical management of the patient was the responsibility of the doctor in charge of cancer treatment, nurses and physiotherapists should provide a non-pharmaceutical response to paralysis, and the liaison team would manage the mental health of the patient and support the mental health of ward staff (Table 1).

#### 2.5.2. Education on How to Deal with Attention Disorders, Rebuilding the Therapeutic Relationship between Patients and Ward Staff: Phase 2

##### Coordination by Liaison Nurse in Phase 2

The liaison nurse suggested how to respond to the concern from the ward staff that resuming SP therapy would worsen the patient’s psychological symptoms. The team worked with the patient and nurses to enable certified psychologists to conduct an evaluation of the patient’s condition using the Japanese version of the Young Mania Rating Scale (YMRS-J) and to consider countermeasures. The liaison nurse assisted in communication about countermeasures to make it easier for the patient and nurses to talk about their concerns. The agreed measures were as follows: (1) the patient should self-check her mental condition; (2) the patient should inform nurses if she becomes aware of any changes in her mental state; (3) nurses should carefully observe any changes in the patient’s mental state; (4) if either the patient or nurses noticed changes in the patient’s mental state, they should discuss this together and consider how to deal with it, and (5) the nurse in charge should inform all ward staff about the coping method jointly decided by the patient and the nurse in charge to provide a unified response (Table 2).

After this discussion, the nurse in charge commented:
“I didn’t point out her psychological symptoms to the patient because I was afraid that doing so would worsen her intimidating attitude, emotional instability, and excessive demands.” She added,
“I learned that the practice of sharing psychological symptoms with patients and considering countermeasures can be an option as an intervention method.”

#### 2.5.3. Managing a Conflict of Views about the Patient’s Discharge from the Hospital: Phase 3

##### Coordination by Liaison Nurse in Phase 3

The liaison nurse suggested that the ward staff should find out more about the background behind the family caregiver’s opposition to the patient’s discharge to the community. She suggested the following three actions: (1) ward staff should try to understand the family caregiver’s concerns; (2) ward staff should identify the reason for concerns and intervene to reduce the family caregiver’s anxiety, and (3) ward staff should guarantee to the family caregiver that the hospital would continue to provide treatment support, psychiatric support, and living support to help the patient use social resources even after she was discharged to the community (Table 3).

## 3. Discussion

In cases where the liaison nurse finds an unrecognized patient with dementia in the ward, a patient has a mental health problem, (e.g., attention disorder or excessive demands) due to dementia caused by chemotherapy and has an issue with the discharge destination, or there is a conflict of intention between the patient and family caregiver regarding the discharge destination, the liaison nurse will coordinate interventions as described below.

Delirium is the most common complication in patients with advanced cancer. However, it is often difficult to identify, which leads to improper management [20]. Delirium in younger patients in particular may be missed because of the lack of disorientation [21]. The patient in this case study showed emotional instability and an intimidating attitude toward the nurses. The patient was in her 40s and the ward staff may have failed to recognize her delirium because she had no symptoms of disorientation. If someone identifies a patient with delirium-induced psychological symptoms that have not previously been recognized, it is important to provide a correct diagnosis as soon as possible, so that appropriate treatment or care can be provided. In this case, the correction using a team approach centered on the liaison psychiatrist helped the ward staff to quickly correct the misrecognition. It seems likely that ward staff will find it easier to accept a correction from the liaison team as a whole, rather than the liaison nurse alone. We suggest that this approach enabled the misrecognition to be corrected promptly. The ward staff and liaison team then correctly diagnosed the situation and worked together to solve the problem.

It is probable that the ward staff had high levels of negative feelings toward the patient because of the stress placed on them by the patient’s intimidating attitude and excessive demands. Nurses should not rush to diagnose a patient with delirium, but should be aware that patients may have this condition [22]. Cases of hypomanic symptoms have been reported during S1 and cisplatin therapy for patients with gastric cancer [23], but it is not common. It is, therefore, possible that the staff on this ward were unfamiliar with psychological symptoms and did not appreciate the possibility of delirium. However, they did recognize the association with the SP therapy and suggested that, if the symptoms recurred on further treatment, it would be difficult for them to manage. The liaison nurse found that the ward staff felt responsible for managing all the patient’s psychological symptoms when they occurred on the ward. The liaison nurse, therefore, contributed to the continuation of treatment by observing the mental health symptoms of the ward staff and working with the patient to enable her to self-check her own symptoms.

In Phase 3, the patient and family caregiver disagreed about discharge. The patient wanted to be discharged to the community, but the family caregiver thought this would be difficult because of the patient’s complex psychological symptoms. Ward staff supported the family caregiver’s view. This is considered inappropriate in this case because they had initially supported discharge and because of the patient’s short life expectancy. It is important to provide information and involve family caregivers when older people are discharged from the hospital [24]. Family caregivers involved in discharge planning are more likely to accept the role of providing post-discharge care [25]. In this case, the family caregiver’s view suggests that they were given little information and not involved in the discharge planning. A previous study found that failure to discharge a COPD patient because of a bureaucratic organizational workflow may not be in the patient’s best interests [26]. In this case, it is possible that ward staff felt that the family caregiver’s opposition to the patient’s discharge to the community would make the process harder. This suggests that the organization as a whole may not be focused on the best interests of the patient. Other studies have noted that the actions of liaison nurses in the discharge process, and especially coordinating with a specialist on behalf of the patient, help to continue care after discharge [27]. In this case, the liaison nurse’s intervention to prompt the ward staff to provide more information to the family caregiver may have contributed to discharge, in line with the patient’s wishes. The liaison nurse asked the ward staff to explain to the family caregiver about the changes in the patient’s personality, and provide more information about the attention disorder, and how it affected behavior. The liaison team also asked the ward staff to arrange for treatment and home-based services for the patient after she was discharged. This support included home-visit nursing services, confirmed contact information in an emergency, a continuous support system by the hospital, and access to a consultation desk for family caregivers. In this case, the liaison nurse helped the ward staff to understand the challenges, and also evaluated their direct care and support to the patient and family.

This case report has some limitations. First, it is a report of a single case and cannot be generalized. Second, the classification of Phases 1–3 is based on our assessment of the patient’s treatment stage, and not on existing theories or protocols. Third, this paper focused on the intervention of the liaison nurse and its effects; however, the intervention of the liaison team as a whole may also have influenced the effects. Despite these limitations, this case report provides useful information for the practice of liaison nurses working with staff who have difficulty responding to changes in personality and diverse psychological symptoms in patients with advanced cancer.

## 4. Conclusions

If a liaison nurse discovers a patient with delirium that has not been recognized on the ward, we recommend that the mistake is pointed out by the whole liaison team, led by liaison psychiatrists. This is because ward staff may be offended by having their mistake highlighted by another nurse, and may not be prepared to consider the possibility of delirium. For mental health problems such as attention disorder and excessive demands caused by delirium arising from chemotherapy, the liaison nurse should support the ward staff and patient to observe each other’s psychological symptoms and work together to build a relationship that can control the symptoms. To enable patients with advanced cancer to make their own decisions about their lives, the liaison nurse needs to help the ward staff to provide family caregivers with information about the medical condition, its expected future course and symptoms, and professional services that are available to help.

## Figures and Tables

**Figure 1 healthcare-10-01335-f001:**
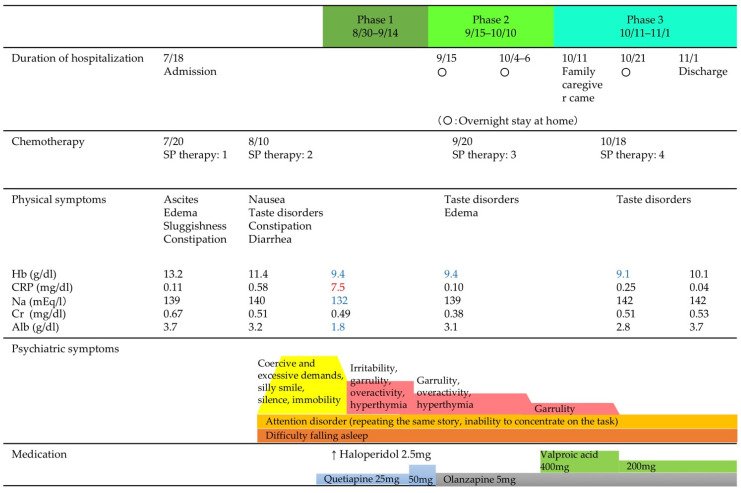
Physical and psychotic symptoms by phase, blood test values, and treatment.

**Table 1 healthcare-10-01335-t001:** Patient’s delirium symptoms in Phase 1, state of ward staff, and liaison nurse intervention needs.

State of Patient	State of Ward Staff	Assessment of Intervention Needs
The patient rapidly developed an intimidating attitude, emotional instability, silly smile, silence, and immobility, and started making excessive demands on the nurses.	Staff decided that the patient’s psychological symptoms were a psychogenic reaction to the news of the cancer. The staff gradually found it more difficult to respond and became more psychologically burdened, and more negative about the patient.	It was difficult for the staff to deal with the patient’s mental health issues because they had failed to recognize that she was experiencing delirium.

**Table 2 healthcare-10-01335-t002:** Patient’s delirium symptoms in Phase 2, state of ward staff, and liaison nurse intervention needs.

State of Patient	State of Ward Staff	Assessment of Intervention Needs
Irritability, attention disorders, and overactivity remained, and the psychological symptoms worsened because of the resumption of SP therapy. Self-care ability therefore declined.	The ward staff decided that it would be difficult to deal with the mental symptoms because of the resumption of SP therapy. They therefore found it difficult to continue treatment of the patient on the general ward.	Support to manage the situation where the ward staff judged that it was difficult to respond to the patient on the general ward.

**Table 3 healthcare-10-01335-t003:** Patient’s delirium symptoms in Phase 3, state of ward staff, and liaison nurse intervention needs.

State of Patient	State of Ward Staff	Assessment of Intervention Needs
The attention disorder and garrulity remained, but the patient expressed her intention of leaving the hospital to live alone in the community.	The ward staff decided that it was possible for the patient to live alone by using social resources such as home-visit nursing. However, the family caregiver was concerned about changes in the patient’s personality, her attention disorder and residual garrulity, and the potential for sudden changes in her illness. He suggested that it would be difficult for her to live alone. The ward staff supported this view once expressed.	The liaison nurse needed to manage the conflict of views about the patient’s discharge from the hospital. This was particularly important given the small amount of time left for her to live, and the potential that her wishes might not be respected.

## Data Availability

Not applicable.

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
