# Peer review of "Intervention of Coordination by Liaison Nurse Where Ward Staff Struggled to Establish a Therapeutic Relationship with a Patient Because of Failure to Recognize Delirium: A Case Study"

_healthcare, 2022, doi:10.3390/healthcare10071335_

Round 1
Reviewer 1 Report
Thaank you for the possibility to review the manuscript entitled: "Intervention of coordination by liaison nurse where ward staff 2 struggled to establish a therapeutic relationship with a patient 3 because of failure to recognize delirium: a case study".
The manuscript is very interesting and more details are considered. However, some minor changes are necessary in order to preceed further for publication.
The CARE guidelines for case report empowered by EQUATOR should be full fill in order to better improve the quality of the manuscript.
References could be updated.
Some typos errors in english language could be revised.
Author Response
添付ファイルをご覧ください。

Reviewer 2 Report
The case report highlights with great detail some of the critical aspects in care provision to patients with a diagnostics of gastric cancer and sheds light on a topic often unaccounted for: delyrium. The authors make a clear case about the added value of liaison nurses in coordinating care provision to these patients and offer valuable insights to ongoing discussion about the quality of care provided to gastric cancer patients.
Author Response
添付ファイルをご覧ください。
